# Octopi-X: Robotic Perception with a Large Tactile-Vision-Language Model for Physical Property Inference

Zexiang Guo[1,*], Hengxiang Chen[1,*], Xinheng Mai[1,*],
Qiusang Qiu[1], Gan Ma[2], Zhanat Kappassov[3], Qiang Li[1,†], Nutan Chen[4]

## I. INTRODUCTION

Accurate perception of object physical properties is fundamental for robots to perform reliable manipulation in unstructured environments. However, current robotic tactile capabilities are still far from matching human touch from different aspects including sensing physical properties [1], response speed and more. Compared to other modalities such as vision, leveraging tactile sensing to improve robotic dexterity are still under exploration. Whereas vision captures geometric structure, it lacks access to underlying mechanical attributes (hardness, elasticity, roughness); by contrast, tactile perception offers this information but necessitates physical interaction, a drawback for delicate or uncertain cases. Recent advances in multimodal learning have shown significant potential for integrating tactile perception with language models to enhance physical reasoning capabilities [2], yet progress is limited by (i) **Sensory constraints in tactile systems:** limited sensory resolution hinders comprehensive characterization of complex materials, and (ii) **Underutilized language model potential:** underexploited language-model reasoning due to suboptimal prompting and fusion. To address these challenges, we propose an enhanced multimodal framework that enable physical property inference for robotic grasping tasks. Our contributions include: (i) **Proactive Perception Architecture:** By fusing visual cues with historical tactile information, our model is capable of predicting important physical attributes. (ii) **Structured Reasoning Prompts:** A staged reasoning protocol that guides multimodal language models through object recognition and property quantification to enhance inference accuracy. (iii) **Instrumented Ground Truth:** Rather than previous work relying on subjective ratings, we use calibrated instruments measuring hardness, elasticity, and roughness for supervision and evaluation. (iv) **Zero-Shot Generalization:** Evaluated on 35 diverse objects, our approach outperforms existing baselines and demonstrates strong zero-shot generalization.

## II. METHODOLOGY

In our method, we introduce a multimodal model integrating textual, visual, and tactile data for comprehensive object

*Equal contribution, †Corresponding author.
[1]School of Artificial Intelligence, Shenzhen Technology University, China, [2]Sino-German College of Intelligent Manufacturing, Shenzhen Technology University, China, [3]Robotics Department, Institute of Smart Systems and Artificial Intelligence (ISSAI), Nazarbayev University, Kazakhstan, and [4]Foundation Robotics, Germany. This research is supported by the "Natural Science Foundation of Top Talent of SZTU"(Grant No. GDRC202411). Also see our project homepage.

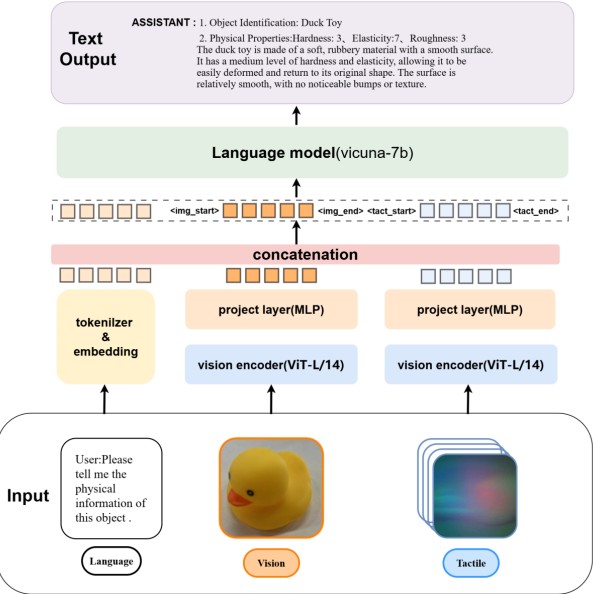

Fig. 1. The architecture of a multimodal large model. After embedding and tokenizing the object image and tactile image alongside the text, the resulting vectors are concatenated and input into the large language model.

analysis. As depicted in Fig. 1, the input query is parsed into dedicated modality-specific pathways. Text is tokenized and embedded via a language tokenizer, while visual and tactile images are encoded using ViT-L/14 [3] and projected into a shared embedding space using modality-specific MLP layers. Special markers (`<img_start>`, `<img_end>`, `<tact_start>`, `<tact_end>`) clearly delineate embedding boundaries. These embeddings are concatenated with textual features and fed into Vicuna-7B [4] , allowing joint multimodal attention to generate detailed object property descriptions, such as hardness, elasticity, and roughness. *Implementation details are provided in the long version (CLAWAR).*

*a) Multimodal Fusion through Feature Concatenation:* After we obtain the projected object image feature vector ($F_o$), the projected tactile image feature vector ($F_t$), and the linguistic feature vector ($F_l$) from the LLM's embedding space, we concatenate them channel-wise into a unified representation:

$$F_{\text{concat}} = [\, F_o \,;\, F_t \,;\, F_l \,].$$

This fused vector $F_{\text{concat}}$ retains distinguishing features from each modality while enabling cross-modal interaction. It then

serves as the input to downstream modules for tasks such as multimodal reasoning, classification, or object recognition, thereby capturing both the physical and semantic attributes of the target object.

*b) Refined Prompting Strategy for Physical Property Scoring:* We designed a structured prompt to enable comprehensive physical property analysis using multimodal (visual and tactile) data. The prompt clearly defines the analysis goal, emphasizing material-aware reasoning and avoiding generic responses. It guides the model through two phases: visual-based object identification (color, shape, texture) and combined material-tactile property evaluation. A 10-point Likert scale quantifies three essential properties, enhancing nuanced differentiation. Outputs include justified object identification and property scores with material rationales. Constraints ensure balanced score usage and material-focused reasoning.

## III. RESULTS

To evaluate our cross-modal perception framework, we conducted comprehensive experiments using a robotic system equipped with a GelSight Mini tactile sensor for high-resolution contact data acquisition and a RealSense D410 camera for visual perception. We selected 35 common household objects spanning diverse materials (plastic, metal, wood, rubber, etc.) and geometric properties. Each object was annotated with ground truth physical properties measured by professional instruments: hardness (Shore scale) with PosiTector SHD, elastic modulus with C610H Auto Tensile Tester, and surface roughness (Ra) with RUGOSURF 20 roughness tester. In additional, we employed designed prompts to assess the physical properties of 35 objects through both our method and the Octopi framework. As can be seen from the Table I, the correlation coefficients between the models and ground truth measurements reveal significant differences in the performance of our model compared to Octopi across the three physical attributes: hardness, elasticity, and roughness (Octopi is the tactile-only model; Octopi-ViTaL is our model).

TABLE I

ZERO-SHOT EVALUATION: COMPARISON OF SPEARMAN'S RANK CORRELATION BETWEEN MODELS AND GROUND TRUTH

| Attribute | Method | Correlation Coefficient | P-value |
|---|---|---|---|
| Hardness | Octopi-ViTaL | **0.501** | **0.005** |
| | Octopi-ViTaL (vision only) | 0.307 | 0.099 |
| | Octopi (fine-grained) | 0.307 | 0.099 |
| | Octopi (original) | 0.015 | 0.935 |
| Elasticity | Octopi-ViTaL | **0.530** | **0.003** |
| | Octopi-ViTaL (vision only) | 0.452 | 0.012 |
| | Octopi (fine-grained) | 0.053 | 0.781 |
| | Octopi (original) | -0.060 | 0.753 |
| Roughness | Octopi-ViTaL | **0.643** | **0.0001** |
| | Octopi-ViTaL (vision only) | 0.413 | 0.023 |
| | Octopi (fine-grained) | -0.010 | 0.959 |
| | Octopi (original) | 0.118 | 0.534 |

**Hardness.** Our model shows a moderate, significant correlation with ground truth ($\rho = 0.501$, $p = 0.005$), out-performing vision-only ($\rho = 0.307$, $p = 0.099$) and both Octopi variants (fine-grained $\approx$ vision-only; original near zero, $\rho = 0.015$, $p = 0.935$).

**Elasticity.** Reporting $|\rho|$ due to the inverse relation to modulus, our approach attains $|\rho| = 0.530$ ($p = 0.003$) vs. vision-only $0.452$ ($p = 0.012$); tactile-only baselines contribute negligible signal (fine-grained $0.053$, $p = 0.781$; original $0.060$, $p = 0.753$).

**Roughness.** Performance separates most clearly: our model reaches $\rho = 0.643$ ($p = 0.0001$) vs. vision-only $0.413$ ($p = 0.023$), while Octopi variants are non-predictive (fine-grained $\rho = -0.010$, $p = 0.959$; original $\rho = 0.118$, $p = 0.534$).

When applied in a zero-shot fashion to our new setup, the pretrained Octopi model failed to produce meaningful predictions (e.g., Spearman's $\rho < 0.1$; see Table I). This failure arises from multiple domain shifts: we use a GelSight Mini with different resolution and calibration compared to Octopi's original high-resolution GelSight; lighting and camera angles differ. These combined shifts in sensor modality, resolution, and lighting prevent Octopi from succeeding zero-shot on our data.

Overall, these results highlight the clear advantage of our multimodal approach. By fusing vision and touch, our model consistently achieves statistically significant and higher correlations with ground truth across all three physical attributes. In contrast, both the vision-only and tactile-only methods—particularly the Octopi framework in its original and adapted forms—fall short, reinforcing the value of cross-modal integration in physical property understanding.

## IV. CONCLUSION

We proposed a novel approach to enhance tactile perception through visual compensation and optimized prompt engineering, leveraging VLM for cross-modal robotic perception. By effectively integrating visual priors and structuring language model interactions, our method overcomes tactile-only limitations and significantly improves physical property inference, especially in roughness estimation. The success of our framework underscores the value of multimodal reasoning with VLMs for robotic applications. Future work will explore applying this multimodal tactile-visual approach to robotic grasping tasks involving adaptive manipulation of objects with different material properties.

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
