# OpenReview forum: "Octopi-X: Robotic Perception with a Large Tactile-Vision-Language Model for Physical Property Inference"
_IEEE.org/IROS/2025/Workshop/Tactile_Sensing — IROS 2025 Workshop Tactile Sensing OralPoster_

### Official Review · Reviewer_hyVm · 2025-09-16
**A insightful work on cross-modal robot perception**

**Rating:** 9
**Confidence:** 4

**Review:**

The author presents a cross-modal model that integrates tactile, visual, and language inputs to infer the physical properties of objects. The work has solid contributions, including a well-structured methodology and a thorough evaluation across a diverse set of real-world objects.

For the camera-ready submission, the reviewer recommends shortening the paper to better fit the workshop format.

---

### Official Review · Reviewer_rpgw · 2025-09-20
**Large Tactile-Vision-Language Model for Physical Property Inference**

**Rating:** 8
**Confidence:** 4

**Review:**

The author uses the visuo-tactile sensing to visualize the tactile information, and then inputs it together with visual information into a large language model to infer physical properties. The results show that this method has high generalization ability. There is a question: How are the two modal features aligned?